# Trait Mindfulness and Problematic Smartphone Use in Chinese Early Adolescent: The Multiple Mediating Roles of Negative Affectivity and Fear of Missing Out

**DOI:** 10.3390/bs13030222

**Published:** 2023-03-03

**Authors:** Yuchang Jin, Wen Xiong, Xinyun Liu, Junxiu An

**Affiliations:** College of Psychology, Sichuan Normal University, Chengdu 610021, China

**Keywords:** trait mindfulness, problematic smartphone use, negative affectivity, fear of missing out

## Abstract

This study used a cross-sectional study design to investigate whether the mindfulness trait was a protective factor against problematic smartphone use (PSPU) of early adolescents, and whether negative affectivity and fear of missing out (FoMO) mediated this relationship. The study selected a sample of middle school students (*N* = 517, 46.03% males, M_age_ = 13.81, SD = 1.40) in China. The results of the structural equation modelling indicated that (a) mindfulness significantly and negatively predicted PSPU, (b) FoMO played a mediating role between mindfulness and PSPU, (c) negative affectivity (including depression and anxiety) played a mediating role between mindfulness and PSPU, but loneliness did not, and (d) negative affectivity and FoMO played a chain-mediated role, and depression, anxiety, and loneliness played a chain-mediated role with FoMO between mindfulness and PSPU. We discuss the possibility that high levels of mindfulness in early adolescents may reduce the short-term effects of problematic smartphone use by reducing negative emotions and FoMO and relate our results to an emphasis on the role of enhanced mindfulness in long-term internal self-regulation and well-being. Findings have implications for individuals and schools for PSPU prevention and intervention.

## 1. Introduction

Smartphones are now an indispensable part of most people’s lives. By December 2018, there were 871 million mobile internet users, with the 20–29 age group accounting for the highest proportion at 26.8% [1]. This is because these smart devices have multiple functions, for example, socializing, entertainment, learning, and online payment. In addition, smartphones have been found to facilitate social communication, enhance emotional connection, improve security, and meet people’s social and emotional needs. There has been an increase in problematic use among young individuals. A survey found that about 50% of American teenagers felt that they had mobile device addiction problems [2]. Research reported that young people show a higher preoccupation with smartphones and are more vulnerable to excessive use [3]. In France, 61% of young people admitted that they were dependent on their smartphones [4]. In China, the rate of mobile phone addiction among teenagers is 10–30%. This percentage is still rising, and the age of smartphone use is getting younger, currently at 11 years [5]. In previous studies, many researchers have focused their research on the negative effects of problematic smartphone use but have rarely examined the antecedents of this phenomenon or focused on the effects of intra-individual factors on PSPU. Under the influence of the epidemic, more work, study, and communication will take place online than ever before. The risk of PSPU may be exacerbated, but there are individual differences. Is this difference influenced by intra-individual factors during adolescence, and are there specific pathways of influence? Mindfulness has been found to reduce problematic smartphone use, but can this effect be applied to early adolescent individuals who are immature, both psychologically and physically?

Researchers claimed that overuse of smartphones could be considered addictive behavior, which features core addictive symptoms (e.g., compulsive behavior, functional impairment, tolerance, and withdrawal), and turned to “addiction” terminology to describe problematic behavior [5,6]. However, some authors believe that there is the possibility of overpathologising [7,8]. As such, this study uses the term problematic smartphone use (PSPU) to refer to maladaptive, uncontrolled, and excessive use of smartphones. Specific characteristics include spending excessive time, money, and resources on the smartphone, constantly checking the smartphone, even in inappropriate situations such as face-to-face communication or driving, and feeling anxious when the smartphone is not around. There are cognitive and socioemotional risk factors associated with PSPU, including social anxiety, low self-esteem, depression, impaired impulse, chronic stress, and poor interpersonal relationships. PSPU has been extensively found to have a negative impact on individual psychosocial adaption and can result in some physical symptoms [9,10,11,12,13]. The World Health Organization [14] also warned that PSPU can have a negative impact. Specifically, PSPU can negatively affect sleep quality, relationships, academic performance, suicidal thoughts, and increased sedentary behavior. Adolescents especially, due to busy learning and school regulations, frequently use smartphones at night, and the electromagnetic field emitted by smartphone will affect the secretion of melatonin, which in turn affects sleep [15]. Adolescents are vulnerable and in a critical period of growth. Therefore, it is of theoretical and practical importance to examine the interplay between the antecedents of problematic smartphone use among Chinese early adolescents. This allows us to have a deeper and more comprehensive understanding of adolescent PSPU and to plan timely, targeted, and effective interventions.

The Interaction of Person-Affect-Cognition-Execution (I-PACE) model [16] pays attention to the interaction between different variables and the psychological and neurobiological mechanisms behind the development of problematic behaviors. I-PACE proposes categories of variables that affect Internet overuse, including predisposing factors and affective and cognitive responses to situational stimuli. I-PACE assumes that response variables contribute to excessive Internet usage, and that affective and cognitive responses (e.g., negative affectivity, FoMO) may mediate between personal characteristics (e.g., trait mindfulness) and problematic use. Based on the I-PACE model, the current study attempts to analyze the relationship between the mindfulness trait and PSPU and analyze systematically the influencing mechanisms in early adolescence, such as the multiple mediation process (i.e., how mindfulness is linked to PSPU).

### 1.1. Mindfulness and PSPU

Individual traits have been found to be robust predictors for PSPU (e.g., neuroticism, extraversion, and poor impulse control) [17] and can also affect PSPU through certain psychological pathways [18]. In recent years, with its protective effects on health receiving extensive attention, mindfulness could be a protective factor to PSPU. Mindfulness as a trait-like construct refers to a state of consciousness that is purposefully and non-judgmentally generated to focus attention on a current experience [19]. It contains two main parts: self-regulation of attention, and an attitude of curiosity, openness, and acceptance. According to “Reperceiving” theory [20], mindfulness raises awareness and cultivates the ability to observe one’s own internal and external experiences. These studies suggest that mindful awareness and attention are associated with positive mental and physical health [21,22]. Targeting adolescents, one study reported a decrease in negative affect and an increase in feeling calm, relaxed, and self-accepting [23]. Automatic and implicit cognitive processes are associated with the maintenance of addictive behaviors. That is, as mindfulness assists people in paying less attention to their emotions and thoughts (e.g., smartphone cravings), automatic maladaptive habits and subsequent dysregulation (e.g., PSPU) are forestalled or diminished [20].

According to Gámez-Guadix and Calvete [24], having a higher level of mindful awareness significantly decreases the probability of problematic Internet use. Other studies have found that mindfulness is inversely correlated with addictive behaviors [25]. Recent studies have found a relationship between mindfulness and mobile phone usage [3,26,27,28]. Mindfulness can reduce behavioral problems in early adolescence [29], such as aggressive behaviors and substance use [30,31]. Mindfulness levels could affect an individual’s development and maintenance of PSPU. However, the possible psychological processes involved have not been clarified. According to the I-PACE theory [16], affective and cognitive responses could mediate between personal characteristics and problematic use.

FoMO and negative affectivity (depression, anxiety, loneliness) are identified as emotional and cognitive factors in the study to explore how mindfulness causes PSPU. This study can support and extend this theory.

### 1.2. Negative Affectivity Mediation

Individuals who have poor emotional control are prone to PSPU [18]. Therefore, negative affectivity, a sign of emotional control failure, could be an important predictor of PSPU. Compensatory Internet Use Theory (CIUT) aims to explain the internal motivation of Internet use, positing that stressful life events have led people to overuse the Internet as a means of alleviating negative emotions. Many studies have also supported the use of CIUTs to explain PSPU [32]. Some studies have found that negative affectivity-related variables, such as anxiety, depression, loneliness, and social anxiety, were associated with excessive smartphone use [6,26,33,34]. In this study, we use negative emotions and FoMO as internal motivating factors of PSPU. Mindfulness is conceptualized as an effective means of emotion regulation. Individuals with high levels of mindfulness were found to remain open, curious, and accustomed to current experiences [35], which promoted positive emotional regulation and reduced negative affectivity [36]. This paper argues that individuals with high levels of mindfulness will better meet their needs and regulate their emotions, thereby reducing PSPU.

This study predicts that one of the key mechanisms may be negative affectivity. It has been found that people who feel lonely have a higher propensity for PSPU [33] and that depression and anxiety are highly correlated with PSPU [6]. But previous studies have mainly focused on adults, and less attention has been paid to early adolescence, which may show different peculiarities due to the emotional instability and cognitive limitations of adolescents. Thus, this study selected these three concepts (depression, anxiety, and loneliness) as the main basis for the evaluation of negative affectivity, which was then taken as the main research variable for the establishment of the mediation model.

### 1.3. FoMO Mediation

A strong desire to communicate and establish relationships with others can motivate individuals to develop dependence symptoms that can lead to PSPU [18]. Social connection has an impact on individuals’ perceived support resources and well-being. Social surveillance theory suggests that individuals can detect social threats and levels of possible rejection, and that FoMO may activate individuals’ social surveillance systems. The fear of missing information about others may thus affect social connections. Therefore, it is proposed that the fear of missing out (FoMO), which is the pervasive apprehension caused by worrying about missing out on rewarding experiences and the desire to keep constant contact with others, could be associated with PSPU. This concern can drive the use of social media to avoid the resulting negative emotion. The behavioral manifestation of FoMO can be observed when people continually participate in social activities or constantly view social media. This strong expectation of wanting to know what others are doing is a cognitive manifestation, whose purpose is to reduce negative emotional states [37].

Recently, there has been an increase in research attention on FoMO, and it has been found that individuals with higher levels of FoMO are more likely to use social media [38,39] and more likely to have social Internet addiction and PSPU [26,40]. From a self-regulation perspective, researchers have seen FoMO as a self-regulatory limbo arising from chronic deficits in psychological need satisfactions [37]. Mindfulness can assist people to concentrate on the present [36] and improve their ability to self-regulate [41], which will be negatively correlated with FOMO. To the best of our knowledge, only one prior study has examined the relationship between mindfulness and FoMO [42]. Based on the above, this study predicts that mindfulness affects adolescent PSPU by influencing FoMO. This is consistent with the I-PACE model, which indicates that the relationship between personal characteristics and problematic behavior is mediated by cognitive factors [16].

In addition, FoMO has been significantly associated with negative emotion. It has been suggested that FoMO could act as a mediator to link deficits in psychological needs with social media engagement [38,43]. In addition, large studies have confirmed the mediating role of FoMO in depression, anxiety, and PSPU severity. Some studies have found that loneliness is associated with boredom tendencies, which may drive FoMO and subsequent mobile phone use, so a new variable of loneliness was added to this study. FoMO is negatively correlated with psychological needs satisfaction; negative affectivity is a manifestation of basic psychological needs frustration. A recent study showed that FoMO mediates negative affectivity and PSPU [44]. According to the I-PACE model, response variables are considered to mediate relations between psychopathology and excessive Internet use, while FoMO is conceptualized as a response variable. Therefore, the present study hypothesized that negative affectivity and FoMO constitute a chain mediation between mindfulness and PSPU.

### 1.4. Current Study

Past research has focused more on cell phone addiction and Internet addiction, and little attention has been paid to problematic smartphone use. Mindfulness has been found to be negatively correlated with PIU, but there are rarely studies that delve into the underlying mechanisms between mindfulness and PSPU [3,26]. In addition, most participants are from the West, with much less focus on Asian groups (especially Chinese groups). Differences between Western and Eastern cultures may cause variables to show different patterns of relationships. The results of previous studies on the relationship between depression, anxiety, and PSPU are relatively consistent, but also found small and opposite relationships. Furthermore, observations on the relationship between PSPU and other relevant psychological constructs were scattered among researchers, with few attempts to investigate these factors simultaneously.

Studies of other psychological structures can help us explain the relationship between mindfulness and PSPU. Based on previous studies and the above theories, we studied the variables of the I-PACE theory, but we also expanded the theory by focusing on other new variables. We believe that in order to comprehensively understand the influence of mindfulness on problematic smartphone use among Chinese early adolescents, we must discuss the mediation mechanism between mindfulness and PSPU. The aims of this study were (a) to test whether mindfulness, as a personal trait, was a protective factor for PSPU in adolescents, and (b) to attempt to understand the mediating mechanisms underlying this relation. To sum up, the main objectives of this research are to have a better understanding of how the mindlessness trait influences the way people are caught in PSPU cycles. This study tested a conceptual model (shown in Figure 1) and put forward the following hypotheses:

**Hypothesis 1:** *Mindfulness negatively predicts PSPU*;

**Hypothesis 2:** *Mindfulness negatively predicts negative affectivity and FoMO*;

**Hypothesis 3:** *Negative affectivity plays a mediating role between mindfulness and PSPU*;

**Hypothesis 4:** *FoMO plays a mediating role between mindfulness and PSPU*;

**Hypothesis 5:** *Negative affectivity and FoMO play a chain mediating role between mindfulness and PSPU*.

## 2. Method

### 2.1. Procedure

The research was approved by the ethics committee of the Sichuan Normal University. Informed consent was obtained from all adolescents included and parents in the study. Prior to the survey, adolescents and parents were provided with information regarding informed consent, and they were told that the purpose of the study was to understand their general smartphone use and their emotional state. First, participants were told that the questionnaire was anonymous and voluntary, and that all data collected were for research purposes only. Then, the willing participants completed the survey independently within the specified time (about 30 min). At the end of the survey, each respondent received a reward.

### 2.2. Participants

Participants were recruited from two junior middle schools in Sichuan, China. A total of 550 questionnaires were distributed, of which 517 were valid (questionnaires with missing values were eliminated), with an effective rate of 94%. The participants ranged in age from 12 to 15, with an average age of 13.81 (*SD* = 1.40). There were 238 male students (46.03%) and 279 female students (53.97%).

### 2.3. Measures

#### 2.3.1. Mindful Attention Awareness Scale, MAAS

The scale compiled by Brown and Ryan [45] was translated and revised by Chen et al. [46]. This scale has been mainly used to measure an individual’s attention and awareness of the present. It has 15 items scored using a Likert scale ranging from “1 = Almost always” to “6 = Almost never,” where the higher the score, the higher the level of mindfulness in daily life. Sample items included “I usually focus on what I want to achieve rather than what I want to do at the moment,” and “I find myself not concentrating on what I am doing.” Studies have shown that the Chinese version of MAAS has good reliability and validity with respect to Chinese adolescents and college students [28,47]. The Cronbach’s alpha coefficient for the present sample was 0.75.

#### 2.3.2. Mobile Phone Addiction Index, MPAI

PSPU was measured using the mobile phone addiction index [47]. The scale mainly measures the physical, psychological, and behavioral symptoms caused by the improper use of smartphones. The scale has 17 items with a 5-level Likert scale ranging from “1 = Never” to “5 = Always,” and four dimensions of inability: the control of cravings (sample items included “users consume a lot of time on smart phones without self-control.” Cronbach’s alpha = 0.80), anxiety and feeling lost (sample items included “choosing to use smart phones to avoid facing the real problems of worry and loneliness.” Cronbach’s alpha = 0.74), withdrawal and escape (sample items included, “individual responses over time after stopping smartphone use.” Cronbach’s alpha = 0.70), and productivity loss (sample items included “Individuals who use their smartphones excessively and are less effective in their studies, work and life.” Cronbach’s alpha = 0.65). Studies have shown that MPAI has good reliability and validity with respect to Chinese adolescents and young people [11,28]. The Cronbach’s alpha coefficient in the present sample was 0.90.

#### 2.3.3. Self-Rating Anxiety Scale, SAS

The SAS developed by Zung [48], which measures the subjective feelings associated with individual anxiety, was used, in which there are 20 items with a 4-level Likert scale from “1 = none or a little of the time” to “4 = most or all of the time,” where the higher the score, the stronger the anxiety. Sample items included “I feel my heart beating fast,” and “I feel scared for no reason.” SAS has already been validated on university and middle school adolescents [49]. The Cronbach’s alpha coefficient in this study was 0.80.

#### 2.3.4. Self-Rating Depression Scale, SDS

This scale developed by Zung [50], which measures the subjective feelings associated with individual depression, was used, on which there were 20 items with a 4-level Likert scale from 1 = “none or a little of the time” to “4 = most or all of the time,” where the higher the score, the stronger the depression. Sample items included “I eat as much as I normally do,” and “I’ve noticed that I’m losing weight.” SDS has already been validated on adolescents [51]. The Cronbach’s alpha coefficient in this study was 0.81.

#### 2.3.5. UCLA Loneliness Scale

The UCLA Loneliness Scale (Version 3) was adopted [52], which has a single-dimensional structure with 20 items scored from 1 (Never) to 4 (Always), where the higher the score, the lonelier the individual. Sample items included “I often feel a lack of friends,” and “I often feel left out.” The scale has already been validated on adolescents [53]. The UCLA scale has also been adapted for Chinese adolescents [54]. The Cronbach’s alpha coefficient was 0.83.

#### 2.3.6. Fear of Missing Out, FoMO

The FoMO scale compiled by Przybylski et al. [37] was translated into Chinese. The scale is a single-factor scale with 10 items that measure an individual’s overall online and offline social interaction FoMO, all of which were scored on a 5-point Likert scale ranging from “1 = Not at all true of me” to “5 = Extremely true of me.” The FoMO scale has been validated in adolescents [55,56]. Sample items included “I often use mobile social media to spend time,” and “I would feel lost if I couldn’t use mobile social media for a few days.” The Cronbach’s alpha coefficient in this study was 0.81.

### 2.4. Analysis

SPSS24.0 and Mplus7.4 were used for analysis. The structural equation model was constructed using Mplus7.4, and the goodness of fit assessed using the root mean square error of approximation (RMSEA, adequate fit between 0.07 and 0.08, excellent fit < 0.07), the comparative fit index, and the Tucker-Lewis Index (CFI, TLI, adequate fit from 0.90 to 0.94, excellent fit > 0.94) [57]. The mediating effects of negative affectivity and FoMO were tested using the bias-corrected percentile bootstrap method, for which bootstrap samples were extracted to obtain the robust standard error and bootstrap confidence interval for parameter estimation. If the 95% confidence interval did not overlap with zero, the results were deemed to be statistically significant [58].

## 3. Results

### 3.1. Common Method Bias

As the measurement results for all variables in this study were based on self-reports, there was a possibility of common method bias. We used several procedures to minimize the threat of method biases [59]. To create psychological separation, we stated that there was no connection between these items; to reduce social desirability effects and to increase accuracy, we received data from the students both with pencil and paper and anonymously and informed them that there were no correct answers to these scales. In addition, to avoid sequential errors in the scales, the order of these scales was randomized. Among the most common problems in the comprehension stage of the response process is item ambiguity. We critically examined these scales, keeping the items concise and specific, and providing examples for some of the more difficult concepts.

Furthermore, before the structural equation modeling, a Harman single factor was used to test for common method bias. The results of the unrotated EFA showed that for all the constructs in this study, the biggest factor explained 33.8% of the variance, which was less than the standard critical value proposed by Podsakoff et al. [59] of 40%. In addition, we controlled variance inflation factors (VIFs) [60]. The VIFs in this study are below the threshold value (i.e., VIFs < 3.3). The highest score was 1.19. Therefore, these results suggest that this model is not affected by common method bias.

### 3.2. Descriptive Statistics and Correlation

The means, standard deviations, and Pearson’s correlation matrix for the primary variables are shown in Table 1. There were significant correlations between all variables. It was found that mindfulness was significantly negatively correlated with FoMO, anxiety, depression, loneliness, and PSPU, and that PSPU, FoMO, anxiety, depression, and loneliness were significantly and positively correlated in pairs (shown in Table 2).

### 3.3. Mediation Using SEM

To further investigate the mediation effects of negative affectivity and FoMO, as well as the mediation roles played respectively by anxiety, depression, and loneliness, this study constructed four models. In the first step, to solve the problem that some scales contain a large number of items, and to stabilize parameter estimates, parceling was used to aggregate items randomly into three parcels within each scale (mindfulness, anxiety, depression, loneliness, and FoMO) [61]. In addition, the anxiety, depression, and loneliness scales were aggregated into three indicators of negative affectivity in M1. Predictor variables were then standardized, with age and sex controlled in each model.

Based on the theoretical hypotheses, Mplus7.4 was used to construct and fit the following four latent structural equation models: for M1, using mindfulness as a predictor and PSPU as a dependent variable, a multiple mediating model of negative affectivity and FoMO in mindfulness and PSPU was constructed (in which negative affectivity pointed to FoMO); for M2, anxiety and FoMO were used as mediators between mindfulness and PSPU (where anxiety points to FoMO); for M3, depression and FoMO were used as mediators between mindfulness and PSPU (where depression points to FoMO); and for M4, loneliness and FoMO were used as mediators between mindfulness and PSPU (where loneliness points to FoMO). All models were found to have good fit, as shown in Table 1.

The mediation effect of these models was further tested using the Bootstrap method, and if the 95% confidence interval did not overlap with zero, the results were deemed to be statistically significant. It was found that FoMO mediated the relationship between mindfulness and PSPU (*β* = −0.18, *SE* = 0.06, 95%CI = [−0.302, −0.066], *p* = 0.002), that the mediating effect of negative affectivity was significant between mindfulness and PSPU (*β* = −0.14, *SE* = 0.04, 95%CI = [−0.222, −0.053], *p* = 0.001), and that negative affectivity and FoMO played a chain mediating role between mindfulness and PSPU (*β* = −0.09, *SE* = 0.02, 95%CI = [−0.133, −0.041], *p* < 0.001).

As anxiety, depression, and loneliness were modeled as the mediators, each of the negative affectivities played a certain role between mindfulness and PSPU: anxiety was found to mediate the relationships, with the chain mediating effect of anxiety and FoMO being significant; depression mediated the relationships, with depression and FoMO also playing a chain mediating role; and while the loneliness intermediary effect between the two was not significant, loneliness and FoMO had a chain mediating role between mindfulness and PSPU. The specific results are shown in Table 3, Table 4 and Table 5.

## 4. Discussion

Western scholars believe that adolescence is the first step towards adulthood. In China, it is usually thought that adolescence begins in junior high school and ends around the age of 18, which is also the end of senior high school [62]. Early adolescence is thought to be the junior high school year [63], which is also the subject group of this study. This stage is historically regarded as a difficult period marked by “storm and stress,” prone to emotional disorders and problem behavior [64], the result of seeking to individuate from parents and often focusing more on peers in the process [65]. This study explored the impact of mindfulness on PSPU in Chinese early adolescents (11–14 years of age) and presented and analyzed the mediating role of negative affectivity and FoMO. The findings fit with Billieux et al.’s theory [18], in the integrative theoretical model, which has three primary pathways to PSPU: excessive reassurance seeking, impulsivity, and extraversion. The mediating effects of depression, anxiety, and loneliness validate these pathways to PSPU. In addition, this result fits with the I-PACE theoretical model proposed by Brand et al. [16]. Finally, according to the conceptualization of PSPU by the Compensatory Internet Use Theory (CIUT), negative emotions are considered to be a precursor to PSPU. In addition, mobile phones can be used as a convenient tool for individuals to maintain social connection. FoMO, defined as the desire to keep in touch with others constantly, may be positively associated with PSPU. The mediating effects of negative emotions and FoMO confirm this reasoning. It was found that mindfulness significantly and negatively predicted PSPU, which supported hypothesis 1. This confirms the findings in Mendelson et al. [66], who reported that mindfulness can increase self-regulation and stress-coping abilities in early adolescents. As mindfulness plays an important role in reducing unconscious thinking and unhealthy behavioral patterns, people with lower mindfulness often unconsciously over-engage with smartphones [3]. It has been found that excessive use of digital technologies can lead to addictive usage patterns [67]. Further, mindfulness can improve early adolescents’ social and emotional skills [66]. In addition, targeting early adolescents can improve the management of attention. Mindfulness can enhance attention and focus on what should be done in the moment [68]; people with high mindfulness tend to be able to focus their attention to the task at hand and avoid excessive mobile phone use, thus avoiding the addictive consequences of overuse. The direct pathway of mindfulness to well-being lies in the fact that individuals with high levels of mindfulness have more high-quality and positive here-and-now experiences, like the flow experience. Self-determination theory suggests that open awareness plays an important role in aligning behavior with an individual’s needs, values, and interests. Mindfulness may enable teenagers to disengage from automatic thoughts, habits, and unhealthy patterns of Internet behavior, which can progress to PSPU. An increase in mindfulness optimizes self-regulation, satisfying teenagers’ basic psychological needs for autonomy, competence, and relationships through self-regulation (in terms of emotions and behavior) and good behavior patterns, promoting individual well-being in the long term.

Hypothesis 2 was also confirmed. The correlation analysis found mindfulness was significantly and negatively correlated with anxiety, depression, loneliness, and FoMO, and further structural equation modeling found that mindfulness significantly negatively predicted negative affectivity and FoMO. Those with higher mindfulness tend to have higher awareness of emotion [19]. In a mindfulness-based intervention of adolescents with depression, one study found a significant reduction in depression and behavioral problems in adolescents [69]. In mindfulness intervention studies of adolescents diagnosed with ADHD, adolescent behavior and attention were found to be improved [70]. Therefore, people with higher mindfulness are less affected by negative affectivity and less likely to experience FoMO. In addition, as this study was the first to prove that mindfulness may be a predictor of FoMO; these results enrich the individual differences found in research pertaining to FoMO.

The results of the mediation mechanism associated with the relationship between mindfulness and PSPU basically supported the other hypotheses (H3, H4, H5). First, it was found that FoMO was a significant mediator between mindfulness and PSPU; that is, mindfulness reduced the risk of PSPU by controlling FoMO. Self-decision theory [71] can assist in better understanding this relationship. It is believed that as FoMO arises from chronic deficits in basic psychological need satisfactions (autonomy, competence, relationship needs), individuals with FoMO seek to meet their basic psychological needs through specific channels or platforms. Phones can be used to meet these autonomy, relationship, and competence needs, which is why people with FoMO may use smartphones to compensate for their need deficits [72], which increases their risk of PSPU. Combined with hypothesis 2, it is believed that people with low mindfulness are often unable to focus their consciousness on their own current experiences [36] and need to follow other people’s lives or novelty events. These ongoing FoMO experiences can contribute to the problematic usage of social networks and their carriers—the smartphone.

Our study also found that negative affectivity was a mediator between mindfulness and PSPU—that is, high levels of mindfulness reduced the risk of PSPU by reducing negative affectivity, which is consistent with the reperceiving theory of mindfulness [20], which claims that through mindfulness, people can disidentify from negative affectivity rather than be immersed in the negative experience, and avoid subsequent dysregulation such as PSPU. While the mediating effect of loneliness between mindfulness and PSPU was not significant, the chain mediating effect of loneliness and FoMO was found to be significant, which indicated that simple loneliness was not the intrinsic mechanism, but when accompanied by FoMO in other experiences or social events, could lead to an increased risk of PSPU. This may be because lonely students tend to compensate in other ways than just using their smartphone. However, when loneliness engenders a strong need for others and results in FoMO, the convenient properties of the smartphone can lead to increased smartphone use.

It was also found that negative affectivity and FoMO played a chain mediating role, which was generally consistent with the I-PACE model, arguing that the relationship between personal characteristics and problematic behavior is mediated by a cognitive response to internal and external stimulus. For early adolescents, mindfulness interventions can improve depression and anxiety [73] and enhance self-management of emotion [74]. Adolescents who have high mindfulness, however, tend to also have an open and inclusive attitude toward life [19], which can effectively reduce the effect of the negative experiences and make them less prone to maladaptive cognition (FoMO). This means they would rely less on their smartphones to relieve stress.

In summary, mindfulness is a resilient factor with respect to negative emotions, FoMO, and PSPU, and negative emotions and FoMO play a mediating role in mindfulness and PSPU. The results of this study provide valuable meaning for the prevention of adolescents’ PSPU, and the individual differences in adolescents in PSPU can be partially explained by the differences in the levels of trait mindfulness. Although the mindfulness trait is relatively stable, we can improve individual state mindfulness through regular meditation and mindfulness interventions. Although mindfulness interventions were originally developed and studied in adult populations, they are increasingly being applied to schools to maintain student mental health and well-being [75,76]. The Developmental Contemplative Science Framework suggests that mindfulness training is particularly effective at specific developmental stages, such as early adolescence. There is a lack of effective interventions related to school-based anxiety, especially test anxiety; lately, however, mindful art making has received more and more attention. Studies have demonstrated the effectiveness of school mindfulness interventions for mental health and well-being in adolescents, and the effectiveness of a mindfulness-based colouring activity for early adolescence test anxiety and state mindfulness [77]. Clinical settings, like those in educational settings, justify optimism for the potential of mindfulness-based therapies to improve child, adolescent, and family functioning [78]. Targeting adolescents at risk of cardiovascular disease [79], breathing meditation also showed greater reductions for 24-h diastolic blood pressure and heart rate. Britton et al. [80] found that mindfulness-based interventions can improve sleep in adolescents, reduce mood problems, and thus reduce substance abuse. The mechanism of the protective effect of mindfulness should be studied further in order to provide better guidance for PSPU intervention.

## 5. Limitations

Like in all research, there are limitations that need to be considered while interpreting the findings and while planning future research. First, the study explored PSPU in early adolescents, which limits the generalizability of the results. Future studies should attempt to randomize the sample and could examine the phenomenon of PSPU at different developmental stages. Second, compared with objectively measured smartphone use, the self-report method may be vulnerable to a social desirability effect, which will affect the accuracy of the data. Future research can be measured through objective smartphone logs. Furthermore, this study focused on the internal factors and mechanisms underlying PSPU during early adolescence. Future research should explore whether there are other variables that play a role in mindfulness and PSPU. Furthermore, as this study focused only on the individual factors related to PSPU, further research is needed to examine the interaction effects between the individual and the environment. Lastly, as this study was a cross-sectional study, it was not possible to assess the causal relationships between the variables, which means that the direction of each variable’s role could not be fully determined; therefore, longitudinal studies need to be conducted to fully investigate the relationships between these variables and enhance their ecological validity. Despite these limitations, this study provides an intra-individual perspective to further examine the antecedents and psychological pathways of PSPU among Chinese early adolescents, which can help families, schools, and individuals to prevent and reduce PSPU. Because mindfulness can be trained through regular meditation, this provides operational methods for individual self-regulation. Future research should expand the factors that influence PSPU to include other intra-individual factors and influences at the in-group and out-group levels.

## 6. Conclusions

In summary, high levels of mindfulness could be a protective factor against PSPU of Chinese early adolescents and could offset PSPU by reducing negative affectivity and FoMO. This study enriches the theoretical PMPU model to a certain extent and provides evidence for the relationship between mindfulness and PSPU and between mindfulness and FoMO. This study also has practical significance for the prevention of, and interventions for, adolescent PSPU. The results suggested that improving mindfulness could reduce the risk of PSPU. Students with low mindfulness should seek to reduce their negative affectivity to avoid the need to seek affective compensation from their smartphone activities. Finally, students with low mindfulness should also seek to reduce their maladaptive cognition (e.g., FoMO) as much as possible to ensure that they avoid an over-reliance on their smartphone.

## Figures and Tables

**Figure 1 behavsci-13-00222-f001:**
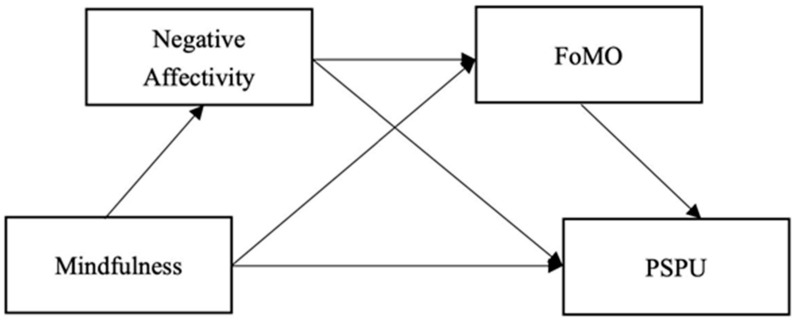
A hypothetical model of the effects of mindfulness on PSPU.

**Table 1 behavsci-13-00222-t001:** The fit indices of each model.

Model	*χ*^2^/*df*	RMSEA	CFA	TLI
M1	3.21	0.08	0.93	0.90
M2	1.98	0.06	0.96	0.94
M3	2.66	0.08	0.93	0.90
M4	2.55	0.07	0.93	0.91

**Table 2 behavsci-13-00222-t002:** Intercorrelations among variables and descriptive statistics.

Variable	*M* ± *SD*	1	2	3	4	5	6
1. Mindfulness	60.01 ± 7.73	1					
2. FoMo	27.44 ± 6.78	−0.20 **	1				
3. Anxiety	36.06 ± 7.84	−0.24 ***	0.47 ***	1			
4. Depression	40.09 ± 7.62	−0.28 ***	0.45 ***	0.71 ***	1		
5. Loneliness	44.86 ± 8.52	−0.28 ***	0.19 **	0.46 ***	0.51 ***	1	
6. PSPU	41.12 ± 10.45	−0.38 ***	0.59 ***	0.45 ***	0.35 ***	0.35 ***	1

Notes: FoMO = Fear of Missing Out; PSPU = Problematic Smartphone Use. ** *p* < 0.01; *** *p* < 0.001.

**Table 3 behavsci-13-00222-t003:** Mediation test, where anxiety was the mediator between mindfulness and PSPU.

Mediating Relationship	*β*	*SE*	95%CI	*p*
Mindfulness- > Anxiety- > PSPU	−0.13	0.04	[−0.208, −0.047]	0.002
Mindfulness- > Anxiety- >FoMO- > PSPU	−0.09	0.03	[−0.142, −0.046]	<0.001
Mindfulness- > FoMO- > PSPU	−0.22	0.07	[−0.350, −0.085]	0.001
Mindfulness- > PSPU	0.19	0.09	[0.019, 0.363]	0.03

Note: PSPU = Problematic Smartphone Use; FoMO = Fear of Missing Out.

**Table 4 behavsci-13-00222-t004:** Mediation test, where depression was the mediator between mindfulness and PSPU.

Mediating Relationship	*β*	*SE*	95%CI	*p*
Mindfulness- > Depression- > PSPU	−0.17	0.05	[−0.277, −0.069]	0.001
Mindfulness- > Depression- >FoMO- > PSPU	−0.08	0.03	[−0.137, −0.018]	0.01
Mindfulness- > FoMO- > PSPU	−0.25	0.07	[−0.397, −0.107]	0.001
Mindfulness- > PSPU	0.25	0.09	[0.187, 0.418]	0.003

Note: PSPU = Problematic Smartphone Use; FoMO = Fear of Missing Out.

**Table 5 behavsci-13-00222-t005:** Mediation test, where loneliness was the mediator between mindfulness and PSPU.

Mediating Relationship	*β*	*SE*	95%CI	*p*
Mindfulness- > Loneliness- > PSPU	−0.01	0.03	[−0.061, 0.082]	0.77
Mindfulness- > Loneliness- >FoMO- > PSPU	−0.09	0.03	[−0.154, −0.031]	0.003
Mindfulness- > FoMO- > PSPU	−0.31	0.08	[−0.476, −0.148]	<0.001
Mindfulness- > PSPU	0.15	0.10	[−0.049, 0.352]	0.139

Note: PSPU = Problematic Smartphone Use; FoMO = Fear of Missing Out.

## Data Availability

The data are available on request to the corresponding author.

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
