# Peer review of "Trait Mindfulness and Problematic Smartphone Use in Chinese Early Adolescent: The Multiple Mediating Roles of Negative Affectivity and Fear of Missing Out"

_behavsci, 2023, doi:10.3390/bs13030222_

Round 1

Reviewer 1 Report

The study is important and valuable, especially in a social context. However, improvements are needed.

Introduction. The authors write that "The rate of mobile phone addiction among Chinese college students is about 10%-40%, and among teenagers is 10%-30%." This percentage is still rising, and the age of smartphone use is getting younger, currently at 10.3 years.” But there is no reference in which sources this information was, because the following reference is from 2018, which is definitely not applicable to "currently". The introduction part must be shortened, saying the main idea in a more concentrated way. At this point, many sentences are repeating themselves, just with slightly different words. And then the Theory section at the end, when everything has been described again. I would recommend integrating this material into the text and shortening it. If the authors study PSPU in teenagers, then it is completely unnecessary to write about driving and road traffic accidents. When writing the Mindfulness and PSPU section, firstly, it should be shortened, and secondly, it is not clear whether the studies mentioned by the authors refer to teenagers, if so, it should be stated. When using the abbreviation for the first time, it should be used in the full text so that it is clear to the reader about what it is about (I-PACE theory), as well as justifying in a few sentences what it has to do with FoMO and negative affectivity. The abbreviation PAPU is also not deciphered. The Negative affectivity mediation section is not necessary or has to be shortened, leaving the idea of why it will be studied and in previous studies that relate to adolescents. At the moment, there is nothing about teenagers. The same applies to the Negative affectivity mediation section. The authors refer to many theories, but it is not clear which one underlies this study. Shorten the Current Study section, especially the beginning.

References in the text should be written the same, according to the requirements, currently the same source has two different notations, for example, Kim, Milne, & Bahl, 2018 and Kim et al., 2018. The authors indicate “shown in Figure 1”, but it does not exist in the text.

Method

Procedure. What is the authors' Institutional Ethics Board? They must name the exact authority that issues the permit and usually it also has a number. It is not clear how the authors ensured the anonymity of the respondents. And given that they are teenagers, parental consent is required for teenagers to be involved in the study. Was it obtained?

Participants. This sentence does not make sense: “A total of 500 questionnaires were distributed, of which 517 were valid…”

Measures. The authors do not indicate whether the questionnaires Self-Rating Depression Scale, Loneliness Scale and Self-Rating Anxiety Scale have been adapted for adolescents, who has done so, where the result is published (reference). The survey Fear of Missing Out, FoMO, is adapted for young adults, but this does not mean that it can be used to survey teenagers without adaptation.

Results.

The authors refer to Table 1, Table 2, 3-5, Figure 2, but there are no such tables and figures. Therefore, it is difficult to evaluate the text.

Discussion.

The entire text, including the introduction part, does not take into account the teenage age, which is different from the adult age, with its own psychological and cognitive peculiarities. Since the theory is not based on adolescent mindfulness research, the text is rather biased. Also, the uses listed by the authors, "Mindfulness-based interventions have been used in various fields,..." do not apply to teenagers, because they have not been conducted in adolescent/teenage samples/selections. I would advise the authors to review the sources used and base their conclusions and discussion on studies of adolescents rather than studies in general, especially since the sample in this study is early adolescence (13 years).

Author Response

Response to Reviewer #1:

We deeply appreciate your valuable suggestions. We have revised the manuscript accordingly.

The comments can be summarized as follows:

  1. Title: Introduction: 1) The authors write that "The rate of mobile phone addiction among Chinese college students is about 10%-40%, and among teenagers is 10%-30%." This percentage is still rising, and the age of smartphone use is getting younger, currently at 10.3 years.” But there is no reference in which sources this information was, because the following reference is from 2018, which is definitely not applicable to "currently".

Response: Dear reviewer, thank you very much for your helpful comments. The "currently at 10.3 years." has been changed to "currently at 11 years." (See Page 1).

  1. Title: Introduction: 1) The introduction part must be shortened, saying the main idea in a more concentrated way. At this point, many sentences are repeating themselves, just with slightly different words.

Response: Thank you for giving us this valuable advice. After reading and searching, the introduction was partially streamlined, some inappropriate and repetitive elements were removed, and some new content was added (See Page 1-2).

  1. Title: Introduction: 1) And then the Theory section at the end, when everything has been described again. I would recommend integrating this material into the text and shortening it.

Response: Thanks for the valuable opinions of the expert. The Theory section at the end in the article has been shortend and integrated into the introduction and discussion (See Page 1 and 10).

  1. Title: Introduction: 1) If the authors study PSPU in teenagers, then it is completely unnecessary to write about driving and road traffic accidents.

Response: Thank you for pointing out this problem. The sentence about ' driving and road traffic accidents.' has been removed.

  1. Title: Introduction: 1) When writing the Mindfulness and PSPU section, firstly, it should be shortened, and secondly, it is not clear whether the studies mentioned by the authors refer to teenagers, if so, it should be stated.

Response: Thank you for giving us such an important advice. Regarding the first question raised, this section removes some worthless content and refines the verbose content by reading and organizing. Regarding the second question raised, Studies on adolescents are clearly pointed out (See Page 1-2).

  1. Title: Introduction: 1) When using the abbreviation for the first time, it should be used in the full text so that it is clear to the reader about what it is about (I-PACE theory), as well as justifying in a few sentences what it has to do with FoMO and negative affectivity.

Response: Thank you for pointing out this problem. The expression “I-PACE theory.” has been changed to “The Interaction of Person-Affect-Cognition-Execution model” in the full text. as for link of FoMO and negative affectivity, the model points that FoMo and negative affectivity as affective and cognitive responses Play mediate role. Also, FoMo as a cognitive response is closely related to negative affectivity (See Page 1- Paragraph 2)).

7 Title: Introduction: 1) The abbreviation PAPU is also not deciphered.

Response: Thank you for pointing out this problem. The expression "PAPU" has been removed.

  1. Title: Introduction: 1) The Negative affectivity mediation section is not necessary or has to be shortened, leaving the idea of why it will be studied and in previous studies that relate to adolescents. At the moment, there is nothing about teenagers. The same applies to the Negative affectivity mediation section.

Response: Thank you for your reminder. After collating the literature, The Negative affectivity mediation section has been shorted, and the shortcomings of previous research are pointed out to put forward the views of this research. Articles about teenagers were clearly identified and new articles about teenagers were added (See Page 3).

  1. Title: Introduction: 1) The authors refer to many theories, but it is not clear which one underlies this study.

Response: Thanks for the valuable opinions of the expert. The Interaction of Person-Affect-Cognition-Execution model has been pointed to underlie this study at the end of introduction section (See Page 1).

  1. Title: Introduction: 1) Shorten the Current Study section, especially the beginning.

Response: Thanks for the valuable opinions of the expert. Through the organization of literature and language, the recent research section has been shorted and some erroneous grammar has been corrected (See Page 4-5).

  1. Title: Introduction: 1) References in the text should be written the same, according to the requirements, currently the same source has two different notations, for example, Kim, Milne, & Bahl, 2018 and Kim et al., 2018.

Response: Thank you for pointing out this problem. By checking, the format of the references in this article has been modified. The expression "Kim, Milne, & Bahl, 2018." has been changed to "Kim et al., 2018" (See Page 4 and 10).

  1. Title: Introduction: 1) The authors indicate “shown in Figure 1”, but it does not exist in the text.

Response: Thank you for pointing out this problem. Figures and tables have been showed in the text (See Page 5, 7, 8 and 9).

Method

  1. Procedure: What is the authors' Institutional Ethics Board? They must name the exact authority that issues the permit and usually it also has a number. It is not clear how the authors ensured the anonymity of the respondents. And given that they are teenagers, parental consent is required for teenagers to be involved in the study. Was it obtained?

Response: Thank you for pointing out this problem. Ethics approval: The research was approved by the ethics committee of the Sichuan Normal University. Consent to participate: Informed consent was obtained from all adolescent participants included in the study and their parents (See Page 5-Paragraph 2).

  1. Participants: This sentence does not make sense: “A total of 500 questionnaires were distributed, of which 517 were valid…” 

Response: Thank you for your reminder, which helped us to clarify the sample in the article and improve the quality of the article. Accordingly, we have revised the "500" to "550" (See Page 5-Paragraph 3).

  1. Measures: The authors do not indicate whether the questionnaires Self-Rating Depression Scale, Loneliness Scale and Self-Rating Anxiety Scale have been adapted for adolescents, who has done so, where the result is published (reference). The survey Fear of Missing Out, FoMO, is adapted for young adults, but this does not mean that it can be used to survey teenagers without adaptation.

Response: Thank you for your valuable suggestion. The questionnaires Self-Rating Depression Scale (Ruiz-Grosso et al., 2012), Loneliness Scale (Kwiatkowska et al., 2017), Self-Rating Anxiety Scale (Shen et al., 2017) and The scale Fear of Missing Out have been adapted for adolescents (Citko & Owsieniuk, 2020; See Page 5).

  1. Results: The authors refer to Table 1, Table 2, 3-5, Figure 2, but there are no such tables and figures. Therefore, it is difficult to evaluate the text.

Response: Thanks for the valuable opinions of the expert. All of figures and tables have been showed in the text (See Page 5, 7, 8 and 9).

  1. Discussion. 1) The entire text, including the introduction part, does not take into account the teenage age, which is different from the adult age, with its own psychological and cognitive peculiarities.

Response: Thank you for your valuable suggestion. At the beginning of the discussion section, the teenage age, with its own psychological and cognitive peculiarities have been discuss based on Literature (See Page 10-Paragraph 1).

  1. Discussion. 2) Since the theory is not based on adolescent mindfulness research, the text is rather biased.

Response: Your suggestion is appreciated, which stimulated us to consult more papers on this topic. Recognizing this problem, I looked up and added many articles on mindfulness interventions for adolescents, summarized their opinions from these articles, and explained them along with the theory (See Page 10 -12).

  1. Discussion. 3) Also, the uses listed by the authors, "Mindfulness-based interventions have been used in various fields," do not apply to teenagers, because they have not been conducted in adolescent/teenage samples/selections. I would advise the authors to review the sources used and base their conclusions and discussion on studies of adolescents rather than studies in general, especially since the sample in this study is early adolescence (13 years).

Response: Thanks for your valuable advice. In order to highlight the effectiveness of adolescent mindfulness interventions, articles on adolescent mindfulness interventions were found and added, and the effectiveness of mindfulness interventions in maintaining mental and physical health was explained from the special group of early adolescents, and then the importance of mindfulness interventions to reduce negative emotions and problem behaviors in the short term and increase early adolescents' well-being in the long term (See Page 11-12 ).

Reviewer 2 Report

Dear Authors,

Thank you for the opportunity to review an interesting article entitled: ‘Trait Mindfulness and Problematic Smartphone Use in Chinese Early Adolescent: The Multiple Mediating Roles of Negative Affectivity and Fear of Missing Out’. The aim of this article is to attempt a systematic analysis of the relationship between the trait of mindfulness and PSPU, as well as mechanisms of influence in early adolescence, such as the process of multiple mediation.

The strengths of the article presented for evaluation are the solid theoretical background and the use statistical analyses. The citation of current literature is also a strong point of this paper.

The reviewer's job, on the other hand, is to help improve the article so that it meets the highest possible standards of the journal, therefore I will focus on its weaknesses.

General comments

[1].  The lack of tables and figures included in the text significantly hinders the analysis of the results and evaluation of the article.

[2].  Minor editorial errors and unintelligible passages are highlighted in yellow in the text.

Abstract

[3].  Use appropriate abbreviations, e.g. Mage instead of mean age, SD instead of standard deviation, etc.

Method

[4].  Incorrectly stated number of questionnaires distributed. Since the final number taken into account in the analysis was 517, 500 could not have been distributed.

[5].  Has the UCLA scale been adapted in China? If so, please provide this information.

Author Response

Response to Reviewer #2

Dear reviewer, thank you so much for your affirmative and helpful comments to this study, which are very significant to improve the quality of the paper.

  1. General comments: 1) The lack of tables and figures included in the text significantly hinders the analysis of the results and evaluation of the article.

Response: Thank you for your valuable suggestion. All of figures and tables have been added in the text (See Page 5, 7, 8 and 9).

  1. General comments: 2) Minor editorial errors and unintelligible passages are highlighted in yellow in the text.

Response: Thank you for your careful reading of our manuscript. Indeed, there are many grammar and spelling mistakes in the paper.Minor editorial errors and unintelligible passages have been modified.

  1. Abstract: Use appropriate abbreviations, e. g. Mage instead of mean age, SD instead of standard deviation, etc.

Response: Thank you for your valuable suggestion. Inappropriate abbreviations has been modified, the expression "mean age" has been changed to "Mage" and the expression "standard deviation" has been changed to "SD" (See page 1- abstract section).

  1. Method: 1) Incorrectly stated number of questionnaires distributed. Since the final number taken into account in the analysis was 517, 500 could not have been distributed.

Response: Thanks for the valuable opinions of the expert. Accordingly, we have revised the "500" to "550" (See Page 5-Paragraph 3).

  1. Method: 2) Has the UCLA scale been adapted in China? If so, please provide this information.

Response: Thanks for the valuable opinions of the expert. UCLA scale has been adapted in Chinese adolescent (Zhang et al., 2015; See Page 6).

Round 2

Reviewer 1 Report

I recommend  o correct some grammar mistakes (Discussion part - first two sentences), also take out sentence "This study was approved by the Human Ethics Committee of Sichuan Normal University." from 2.2 part, because it is in 2.1 part.

Author Response

I recommend o correct some grammar mistakes (Discussion part - first two sentences), also take out sentence "This study was approved by the Human Ethics Committee of Sichuan Normal University." from 2.2 part, because it is in 2.1 part.

Response: Thanks for your valuable suggestion. We have corrected some grammar mistakes in discussion part. We take out sentence “This study was approved by the Human Ethics Committee of Sichuan Normal University." from 2.2 part.

Again, we are very grateful for the reviewers’ insightful comments and suggestions, which are extremely valuable and helpful in improving the quality of our manuscript.

Reviewer 2 Report

Dear Authors,

The paper submitted for re-evaluation entitled: 'Trait Mindfulness and Problematic Smartphone Use in Chinese Early Adolescent: The Multiple Mediating Roles of Negative Affectivity and Fear of Missing Out' still contains several errors:

[1].  Minor editorial errors and unintelligible passages are highlighted in yellow in the text as are missing bibliographic entries and unquoted items.

[2].  The notation of the 'p' designations, which should be in italics in each case, should be standardised.

[3].  The formatting of all text should be checked so that it is uniform.

[4].  Lack of consistency in literature citation: once all authors are listed e.g. (Kim, Milne, & Bahl, 2018) and other times only the first author is listed e.g. (Richter et al., 2022) despite the fact that in both cases there are three authors each.

[5].  "The FoMO has been validated in adolescents (Citko & Owsieniuk, 2020)." - this is not true, Citko & Owsieniuk developed their own FoMO index based on the Przybylski et al. scale.

[6].  Both the references to the bibliography and the bibliography itself are not prepared in accordance with the requirements of the journal - please take a look at the information at https://www.mdpi.com/journal/behavsci/instructions#preparation.

Author Response

  1. Minor editorial errors and unintelligible passages are highlighted in yellow in the text as are missing bibliographic entries and unquoted items.

Response: Thanks for your valuable marks. The highlighted minor editorial errors and unintelligible passages in the text have been identified and modified.

  1. The notation of the 'p' designations, which should be in italics in each case, should be standardised.

Response: Thank you for your careful reading of our manuscript. The notation of the 'p' designations has been modified to italics (See Page 7-8).

  1. The formatting of all text should be checked so that it is uniform.

Response: Thank you for your reminder. Through inspection, some inconsistent expressions in the text were changed to consistent expressions, and the content of change is identified.

  1. Lack of consistency in literature citation: once all authors are listed e.g. (Kim, Milne, & Bahl, 2018) and other times only the first author is listed e.g. (Richter et al., 2022) despite the fact that in both cases there are three authors each.

Response: Thank you for your reminder. Through a one-to-one correspondence check of the reference, those inconsistencies were modified to express consistently.

  1. "The FoMO has been validated in adolescents (Citko & Owsieniuk, 2020)." - this is not true, Citko & Owsieniuk developed their own FoMO index based on the Przybylski et al. scale.

Response: Your suggestion is appreciated. By reading carefully, we looked for other papers where the scale was applied to adolescents and replaces the previous article.

  1. Both the references to the bibliography and the bibliography itself are not prepared in accordance with the requirements of the journal - please take a look at the information at https://www.mdpi.com/journal/behavsci/instructions#preparation.

Response: Thanks for your valuable suggestion. By reading carefully, the formatting in the text has been modified according to the format required by the journal.

Again, we are very grateful for the reviewers’ insightful comments and suggestions, which are extremely valuable and helpful in improving the quality of our manuscript.
